# NEURAL NETWORKS WITH MOTIVATION

## ABSTRACT

How can animals behave effectively in conditions involving different motivational contexts? Here, we propose how reinforcement learning neural networks can learn optimal behavior for dynamically changing motivational salience vectors. First, we show that Q-learning neural networks with motivation can navigate in environment with dynamic rewards. Second, we show that such networks can learn complex behaviors simultaneously directed towards several goals distributed in an environment. Finally, we show that in Pavlovian conditioning task, the responses of the neurons in our model resemble the firing patterns of neurons in the ventral pallidum (VP), a basal ganglia structure involved in motivated behaviors. We show that, similarly to real neurons, recurrent networks with motivation are composed of two oppositely-tuned classes of neurons, responding to positive and negative rewards. Our model generates predictions for the VP connectivity. We conclude that networks with motivation can rapidly adapt their behavior to varying conditions without changes in synaptic strength when expected reward is modulated by motivation. Such networks may also provide a mechanism for how hierarchical reinforcement learning is implemented in the brain.

## 1 INTRODUCTION

Motivation is a cognitive process that propels an individual's behavior towards or away from a particular object, perceived event, or outcome (Zhang et al., 2009). Mathematically, motivation can be viewed as subjective modulation of the perceived reward value before the reward is received. Therefore, it reflects an organism's wanting of the reward before the outcome is actually achieved.

Computational models for motivated behavior, which are best represented by reinforcement learning (RL) models, are mostly concerned with the learning aspect of behavior. However, fluctuations in physiological states, such as confidence and motivation, can also profoundly affect behavior (Zhang et al., 2009). Modeling such factors is thus an important goal in computational neuroscience and is in the early stages of mathematical description (Berridge, 2012).

Here we build a neural network theory for motivational modulation of behavior based on Q-learning and apply this theory to mice performing Pavlovian conditioning task in which experimental observations of neural responses obtained in the ventral pallidum (VP) are available. We show that our motivated RL model both learns to correctly predict motivation-dependent rewards in the Pavlovian conditioning task and is consistent with responses of neurons in the VP. In particular, we show that, similarly to the VP neurons, Q-learning neural networks contain two oppositely-tuned populations of neurons responsive to positive and negative rewards. In the model, these two populations form a push-pull network that helps maintain motivation-dependent variables when inputs are missing. Our RL-based model is both consistent with experimental data and predicts the structure of the VP networks. We thus argue that motivation leads to complex behaviors which may add an extra level of complexity to machine learning approaches and is consistent with biological data.

## 2 RESULTS

Motivation is defined mathematically as a need-dependent modulation of the perceived reward value depending on animal's extrinsic or intrinsic conditions (Zhang et al., 2009). Thus, rats, which are normally repelled by high levels of salt in their food, may become attracted to a salt-containing solution following salt-free diet (Berridge, 2012). To model this observation, Berridge & Schulkin

(1989) have proposed that the perceived reward $r_t$ received at time $t$ is not absolute, but is modulated by an internal variable reflecting the level of motivation, which we will call here $\mu$. The perceived level of the reward $\tilde{r}_t$ as a function of motivation $\mu$ can be expressed by the following equation:

$$\tilde{r}_t = \tilde{r}(r_t, \mu) \tag{1}$$

In the simplest example, the reward, associated with salt is given by $\tilde{r}_t = \mu r_t$. Baseline motivation towards salt can be defined by $\mu = -1$, leading to the perceived reward of $\tilde{r}_t = -r_t < 0$. Thus, normally the presence of salt in the diet is undesired. In the salt-free condition, the motivation changes to $\mu = +1$, leading to the subjective reward of $\tilde{r}_t = +r_t \geq 0$. Thus salt-containing diet becomes attractive. In reality, the function $\tilde{r}(...)$ defining the impact of motivation on a perceived reward is complex (Zhang et al., 2009), including the dependence on multiple factors described by a motivation vector $\vec{\mu}$. Individual components of this vector describe various needs experienced by the organism, such as thirst (e.g. $\mu_1$), appetite ($\mu_2$), etc. In this study, we explore the computational impact of motivation vector in the context of RL and investigate the brain circuits that might implement these computations.

Our approach to motivation is based on Q-learning (Watkins & Dayan, 1992), which relies on an agent estimating Q-function, defined as the sum of future rewards given an action $a_t$ chosen in a state $\vec{s}_t$ at time point $t$: $Q(\vec{s}_t, a_t) = \sum_{\tau=0}^{\infty} r(\vec{s}_{t+\tau}|a_t)\gamma^{\tau}$ (here and below, we omit averaging for simplicity). Here $0 < \gamma \leq 1$ is the discounting factor that keeps the sum from diverging, and balances preference of short- versus long-term rewards. If a correct Q-function is known, a rational agent picks an action that maximizes future rewards: $a_t \leftarrow argmax_a Q(\vec{s}_t, a)$. In case of motivation in equation 1, as reward values are affected by the motivation vector $\vec{\mu}$, for the Q-function, we obtain:

$$Q(\vec{s}_t, a_t, \vec{\mu}) = \sum_{\tau=0}^{\infty} \tilde{r}(\vec{s}_{t+\tau}, \vec{\mu}_{t+\tau}|a_t)\gamma^{\tau} \tag{2}$$

Here $\tilde{r}(\vec{s}_{t+\tau}, \vec{\mu}_{t+\tau}|a_t)$ is the motivation $\vec{\mu}$-dependent perceived reward obtained in a state $\vec{s}_{t+\tau}$ reached at time $t + \tau$ given action $a_t$ chosen at time $t$.

The state of the agent $\vec{s}_t$ and its motivation $\vec{\mu}$ are distinct. The motivation is a slowly changing variable, that on average is not affected substantially by a single action. For example, the animal's appetite does not change substantially during a single trial. At the same time, the actions selected by the animal lead to immediate changes of the animal's state $\vec{s}_t$. Recent research in neuroscience suggests that motivation and state may be represented and computed separately in the mammalian brain. Whereas motivation is usually attributed to the regions of the reward system, such as the VP (Berridge & Schulkin, 1989; Berridge, 2012), the state is likely to be computed elsewhere, e.g. in the hippocampus (Eichenbaum et al., 1999), or cortex. In RL, an agent's state and motivation may have different mathematical representations. In the examples below, the state variable is given by a one-hot vector, while motivation is represented by a full vector. Two arguments of the Q-function, $\vec{s}_t$ and $\vec{\mu}$, are therefore distinct. Finally, in hierarchical RL implementation, motivation is provided by a higher level network, while information about the state is generated externally.

Although the Q-function with motivation (equation 2) is similar to the Q-function in goal-conditioned RL (Schaul et al., 2015; Andrychowicz et al., 2017), the underlying learning dynamics is different. Motivated behavior pursues multiple distributed sources of dynamic rewards. The Q-function therefore accounts for the future motivation dynamics. This way, an agent with motivation chooses what reward to pursue – making it also different from RL with subgoals (Sutton et al., 1999). Behavior with motivation therefore involves minimum to no handcrafted features, which suggests that motivation could provide a step towards general methods that leverage computation – a goal identified by Richard Sutton (2019).

As in the case of standard Q-learning, the action chosen by a rational agent maximizes the sum of the expected future perceived rewards, i.e. $a_t \leftarrow argmax_a Q(\vec{s}_t, a, \vec{\mu})$. To learn a correct Q-function, one can use the Temporal Difference (TD) method (Sutton & Barto (1998)). If the Q-function is learned perfectly, it satisfies the recursive relationship $Q(\vec{s}_t, a_t, \vec{\mu}) = \tilde{r}(\vec{s}_t, \vec{\mu}_t) + \gamma \max_{a_{t+1}} Q(\vec{s}_{t+1}, a_{t+1}, \vec{\mu}_{t+1})$. For an incompletely learned motivation-dependent Q-function, the TD error $\delta$ is non-zero:

$$\delta = \tilde{r}(\vec{s}_t, \vec{\mu}_t) + \gamma \max_{a_{t+1}} Q(\vec{s}_{t+1}, a_{t+1}, \vec{\mu}_{t+1}) - Q(\vec{s}_t, a_t, \vec{\mu}_t) \tag{3}$$

TD error can be used to update motivation-dependent Q-function directly or to train neural networks to optimize their policy. Q-function depends on the new set of variables $\vec{\mu}$ that evolve following

their own rules. These variables reflect fluctuations in physiological or psychological states that substantially change the reward function and, therefore, can generate flexible behaviors dependent on animals' ongoing needs. We trained neural networks via backpropagation of the TD error (equation 3), an approach employed in deep Q-learning (Mnih et al., 2015). Below we present several examples in which neural networks could be trained to solve motivation-dependent tasks.

## 2.1 THE FOUR DEMANDS TASK

Consider the example in Figure 1. An agent navigates in a 6x6 square gridworld separated into four 3x3 subdivisions (rooms) (Figure 1A). The environment was inspired by the work of Sutton et al. (1999); however, the task is different, as described below. In each room, the agent receives one and only one type of reward $r_n(x_t, y_t)$, where $n = 1...4$ (Figure 1B). These rewards can be viewed as four different resources, such as water, food, sleep, and work. Motivation is described in this system by a 4D vector $\vec{\mu}$ defining affinity of the agent for each of these resources. When the agent enters a room number $n$, the corresponding resource in the room is consumed, the agent receives rewards defined by $\tilde{r}_t = \mu_n$, and the corresponding component of the motivation vector $\mu_n$ is reset to zero (Figure 1C). On the next time step, motivations in all four rooms are increased by one, i.e. $\mu_n \leftarrow \mu_n + 1$, which reflects additional "wanting" of the resource induced by the "growing appetite". After a prolonged period of building up appetite, the motivation towards a resource saturates at a fixed maximum value of $\theta$, which becomes a parameter of this model, determining the behavior.

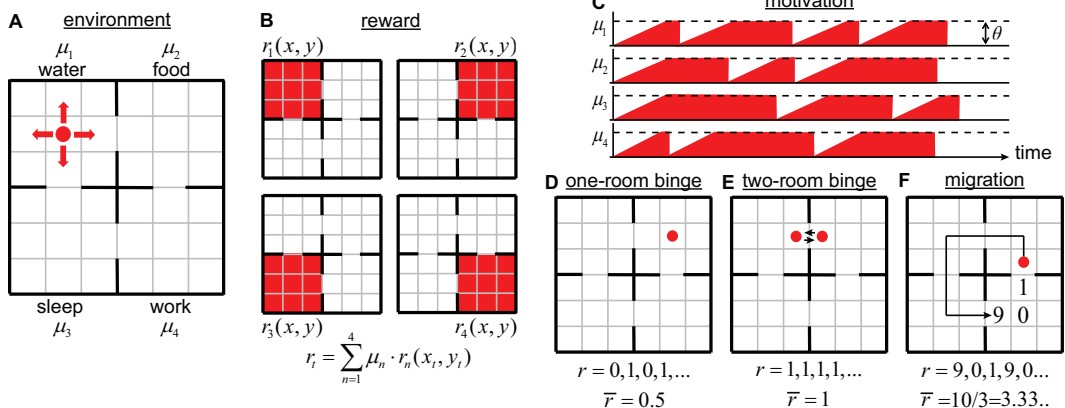

Figure 1: The Four Demands example. (A) An agent populates a 6x6 environment separated into four rooms. Each room is associated with its own reward and motivation (water, food, sleep, work). (B) The perceived reward is a scalar product between the motivation vector and the reward vector as illustrated. (C) Components of the 4D motivation vector as functions of time. When agent enters a room, its motivation is reset to zero. When the agent is not in the room, the motivation increases by 1 at each time step until saturation at $\theta$. (D-F) Potential strategies in our model: (D) one-room binge, (E) two-room binge, and (F) migration.

What are the potential behaviors of the agent? Assume, that the maximum allowed motivation $\theta$ is large, and does not influence our results. If the agent always stays in the same room (one-room binge strategy, Figure 1D), the rewards received by the agent consist of a sequence of zeros and ones, i.e. 0, 1, 0, 1, ... (in our model, after the motivation is set to zero, it is increased by one on the next time step). The average reward corresponding to this strategy is therefore $\bar{r}_{one-room\ binge} = 1/2$. The average reward can be increased, if the agent jumps from room to room on each time step (a two-room binge strategy, Figure 1E). In this case, the sequence of rewards received by the agent is described by the sequence of ones and the average reward is $\bar{r}_{two-room\ binge} = 1$. Two-room binging therefore outperforms the one-room binge strategy. Finally, the agent can migrate by moving in a cycle through all four rooms (Figure 1F). In this case, the agent spends three steps in each room and the overall period of migration is 12 steps. During these three steps, the agent receives the rewards of 9 (the agent left this room nine steps ago), then 0, and 1 ($\bar{r}_{migration} = 10/3$). Thus, migration strategy is more beneficial for the agent than both of the binging strategies. Migration, however, is affected by the maximum allowed motivation value $\theta$. When $\theta < 9$, the benefits of migration strategy are reduced. For $\theta = 1$, for example, migration yields the reward rate of just $\bar{r}_{migration}|_{\theta=1} = 2/3$, which is below the return of the two-room binging. Thus, our model should display various behaviors depending on $\theta$.

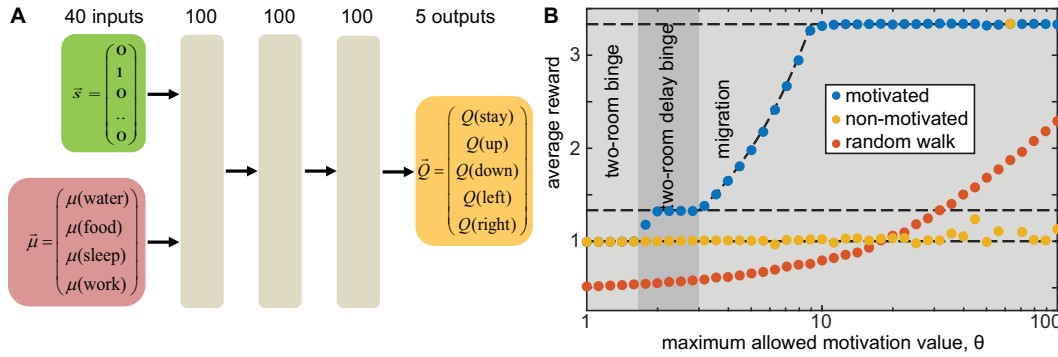

Figure 2: (A) The architecture of the 3-layer fully connected network computing the Q-function $Q(a|\vec{s}, \vec{\mu})$. (B) The average reward per time step received by the network trained with maximum allowed motivation value $\theta$ (blue circles – motivation is provided as an input to the network; yellow – no motivation; orange – random walk). For small $\theta$s, the motivated network displays two-room binge behavior, while for larger $\theta$s, migration dominates. Under the same conditions, the non-motivated network mostly displays two-room binge behavior.

We trained a simple feedforward neural network (Figure 2A) to generate behaviors using the state vector and the 4D vector of motivations as inputs. The network computed Q-values for five possible actions (up, down, left, right, stay), using TD method and backpropagating the $\delta$ signal. The binary 36D (6x6) one-hot state vector represented the the agent's position. The network was trained 41 times for different values of the maximum allowed motivation value $\theta$. As expected, the behavior displayed by the network depended on this parameter. The phase diagram of the agent's behaviors (Figure 2B, blue circles) shows that the agent successfully discovered the migration strategy and two-room binge strategies for high and low values of $\theta$ correspondingly. For intermediate values of $\theta$ $(1.7 < \theta < 3)$, the network discovered a delayed two-room binging strategy, in which it spent an extra step in one of the room. The networks with motivation can also display a variety of complex behaviors for different motivation dynamics, such as binging, addiction, withdrawal, etc. In one example, by increasing the maximum motivatiuon value for one of the demands ("smoking"), we trained networks to display "smoking addiction" (Figure 3A,B).

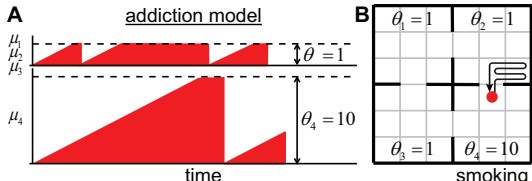

Figure 3: Addiction model: (A) addiction motivation schedule. In the first three rooms, motivations are allowed to grow up to the value of 1. In the fourth ("smoking" room), the motivation may grow to the value of 10. (B) Strategy learned by the network: spend 5 steps in the room #2, then visit the room #4 ("smoking")

Does motivation contribute to learning optimal strategies? To address this question, we performed a similar set of simulations, except the motivation input to the network was suppressed ($\mu = 0$). Although the input to such "non-motivated" networks was sufficient to recover the optimal strategies, in most of the simulations the agents exercised two-room binging (Figure 2B, yellow circles). The migration strategy, despite being optimal in 3/4 of the simulations, was successfully learned only by a single agent out of 41. Moreover, the performance of the non-motivated networks often yielded that of the random walk (Figure 2B, orange circles). We conclude that motivation may facilitate learning by providing additional cues for temporal credit assignment in the rewards. Overall, we suggest that motivation is helpful in generating complex ongoing behaviors based on simple conditions.

## 2.2 THE TRANSPORT NETWORK TASK

In the next example, the agent navigates in a system of roads connecting $N$ cities (Figure 4A). The goal of the agent is to visit a certain subset of the target cities. The visiting order is not important, but the agent is supposed to use the route of minimal length. This problem is similar to the vehicle routing problem (Dantzig & Ramser, 1959) (we do not require agents to return to the city of origin).

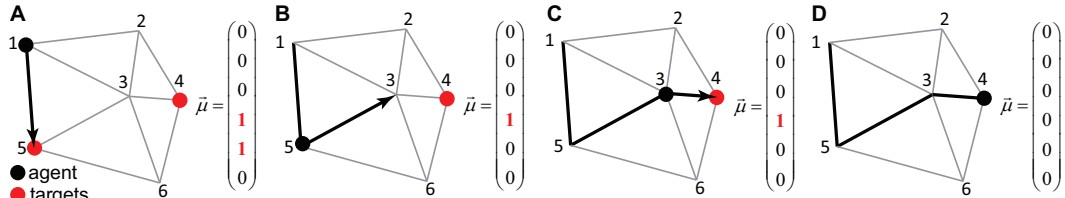

Figure 4: The transport network example. An agent (black dot) navigates in a network of roads connecting the cities – each associated with its own binary motivation. The perceived reward is equal to the value of the motivation vector $\vec{\mu}$ at the position of the agent, less the distance traveled. When the agent visits a city with non-zero motivation (red circle), the motivation toward this city is reset to zero. The task continues until $\vec{\mu} = \vec{0}$. (A-D) The steps of the agent through the network (black arrows) and the corresponding motivation vectors.

We trained a neural network that receives the agent's state (position) and the motivation vector as inputs, then computes the Q-values for all available actions (connected cities) for the given position (Figure 5A). In every city, the agent receives a reward equal to the value of the motivation vector at the position of the agent. The network is also negatively rewarded at every link between cities in proportion to the length of this link. We trained the network using TD method by backpropagating the TD error. Trained neural networks produced behaviors that closely match the shortest path solution (Figure 5B). In 82% of the test examples, the agent traveled the shortest path. In the remaining cases, the paths chosen by agents are close to the shortest path solution. Overall, we suggest that networks with motivation can solve complex transport problems. In doing so, the agent is not instructed to perform any particular goal, but instead learns to set next target autonomously.

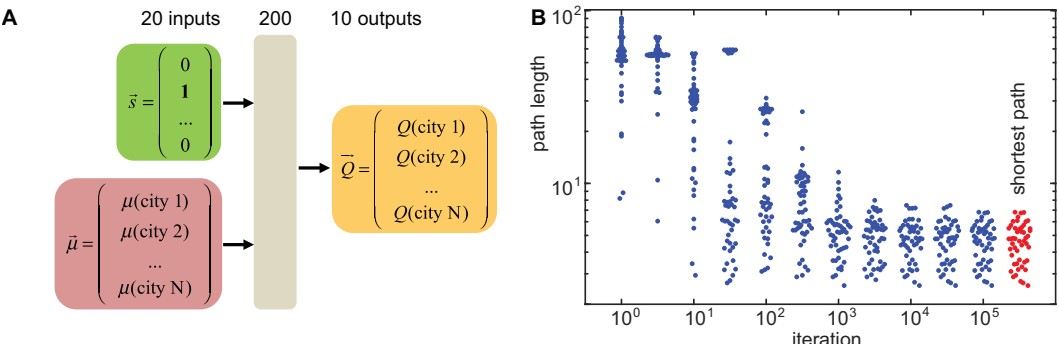

Figure 5: Training a neural networks to find the shortest route to visit a subset of target cities using the motivation framework. (A) One-layer neural network computing the Q-values. (B) Test performance on 50 sets of 3 random targets (blue). The precomputed shortest path solutions (red). The policy converges on iteration $3 \cdot 10^3$.

## 2.3 RESPONSES OF THE VP NEURONS IN PAVLOVIAN CONDITIONING TASK

To explore how motivation may be implemented in the brain, we trained 3 mice to associate the specific cues (sound tones) with the different rewards (Figure 6A,B). In the experiment, the animals received one of five possible rewards: a large or small positive reward (a drop of water); a large or small negative reward (an air puff); or a zero reward – nothing at all. Trials containing positive or negative rewards combined with zero reward trials were separated into different blocks. During these blocks of trials, the animal was expected to be motivated and demotivated respectively. In course of the training, the animals learned to anticipate both positive and negative rewards.

To relate behavior to the underlying neuronal circuits, we recorded the activity of the neurons in the VP – a brain area implicated in computing motivation (Berridge & Schulkin, 1989). The recordings were made while the mice were performing this task (Figure 6A,B). Overall, we obtained 149 well-isolated single neurons that showed task-related responses (Figure 6C). Our data suggests that the VP contains 2 large populations of oppositely-tuned neurons, activated by positive and negative (Figure 6D,E) rewards. To gain insight into a potential explanation for this phenomenon, we investigated artificial neural networks with motivation that were subjected to similar conditions as mice.

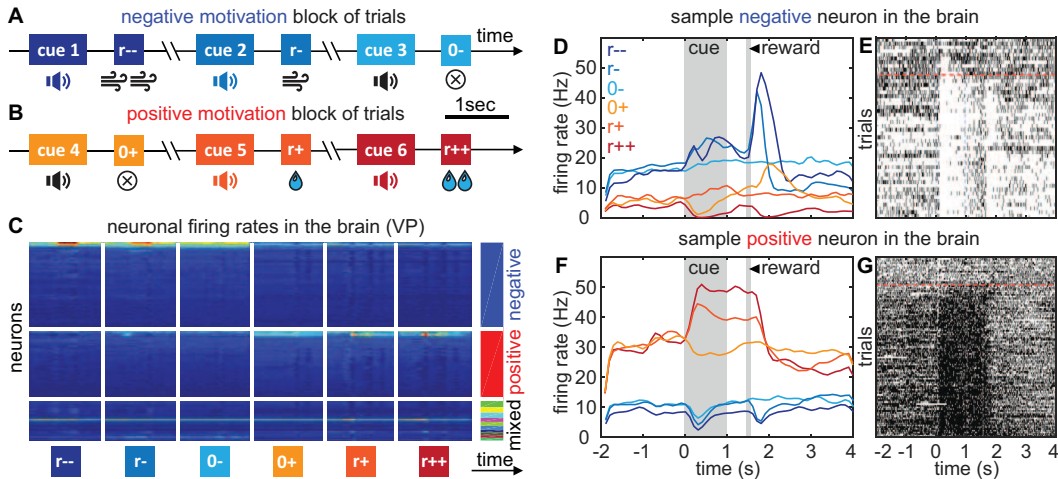

Figure 6: (A, B) A behavioral task for the recording. Trials were separated into the blocks during which only negative rewards (air puffs) or only positive rewards (water drops) were delivered. (C) Responses of neurons recorded in 3 mice clustered into the negative motivation, positive motivation, and neurons of mixed sensitivity. (D-G) Activity of sample neurons responding to negative (D, E) or positive (F, G) rewards.

As the Pavlovian conditioning task includes time as variable (Figure 6), we chose to use recurrent neural network (RNN) as a basis of our model, as suggested by Sutton & Barto (1987). The RNN received 2 inputs. One input described the cue as a function of time within a trial (Figure 7A,B) – representing the state of the animal. Another input described motivation (constant within the entire trial) to indicate whether an agent is in a positive ($\mu = +1$) or negative ($\mu = -1$) block of trials.

The network has learned to accurately predict the trial outcome based on the cue (Figure 7B). For example, in the negative block of trials ($\mu = -1$), before a cue is presented ($s = 0$), the expected value of future reward $V_t(\mu_t, s_t)$ starts from a low negative value, in an expectation of future negative reward. As the cue arrives, the expected value of future reward $V_t$ represents the expected outcome. For example, in the trials with large negative reward (the leftmost column in Figure 7B), the network adjusts its expectation to lower value after the cue arrives ($s = -0.8$). For trials with small negative reward (second column), no adjustment is necessary, and, therefore, reward expectation $V_t$ remains unaffected by the cue. $V_t$ decreases slightly after the cue arrives due to the temporal discount $\gamma = 0.9$. For no-negative-reward trials (Figure 7B, column 3), in the negative block of trials, the expected reward increases after the cue arrives, due to the optimistic prediction. In positive block of trials ($\mu = +1$, Figure 7B, columns 4-6), the behavior of the network is the same, except for the sign. Overall, our model yields reward expectations $V_t$ that accurately reflect motivation and future rewards.

We then examined the responses of neurons in the model. We clustered the responses using unsupervised clustering algorithm (Sinakevitch et al., 2018). The neural population contained two large groups of oppositely tuned cells (Figure 7C), elevating their activity in positive and negative reward trials respectively, in agreement with the experimental observations in the brain (Figure 6C). Overall, we find a close correspondence between activity of neurons in the artificial and biological networks.

What might be the functional significance of the two oppositely tuned neural populations? We found that the negative reward neurons (Figure 7D, blue cluster) tend to form excitatory connections with each other, and so do the positive reward neurons (red cluster). Oppositely tuned cell, on the other hand, tend to inhibit each other (Figure 7E,F). Thus, RNN in our model yields a prediction for the structure of connectivity in the VP in the brain. Such connectivity helps maintaining the information about reward expectation within the trial. Indeed, in the Pavlovian conditioning task, cue and reward are separated by a temporal delay. During the delay, the network is supposed to maintain the information about upcoming reward, and, thus, acts as a working memory network (Her et al., 2016), which keeps reward expectation in its persistent activity. This persistent activity can be seen in both the responses of individual neurons in the VP in the brain (Figure 6C-E) and the RNN neurons in the model (Figure 7C). Previous studies in working memory and decision-making tasks (Machens et al., 2005; Wong et al., 2007; Her et al., 2016) suggest that such parametric persistent activity can be maintained by two groups of oppositely tuned neurons, in the network architecture called

the "push-pull" circuit. This is exactly what we find in our RNN (Figure 7F). Memory is maintained in push-pull circuits via positive feedback. The positive feedback is produced by two forms of connectivity. First, similarly tuned neurons excite each other, as in Figure 7D. Second, oppositely tuned neurons inhibit each other, which introduces effective self-excitation via disinhibition. Overall, we show that, similarly to real neurons, recurrent networks with motivation are composed of two oppositely-tuned classes of neurons, responding to positive and negative rewards. Our model also generates predictions for the structure of the VP connectivity.

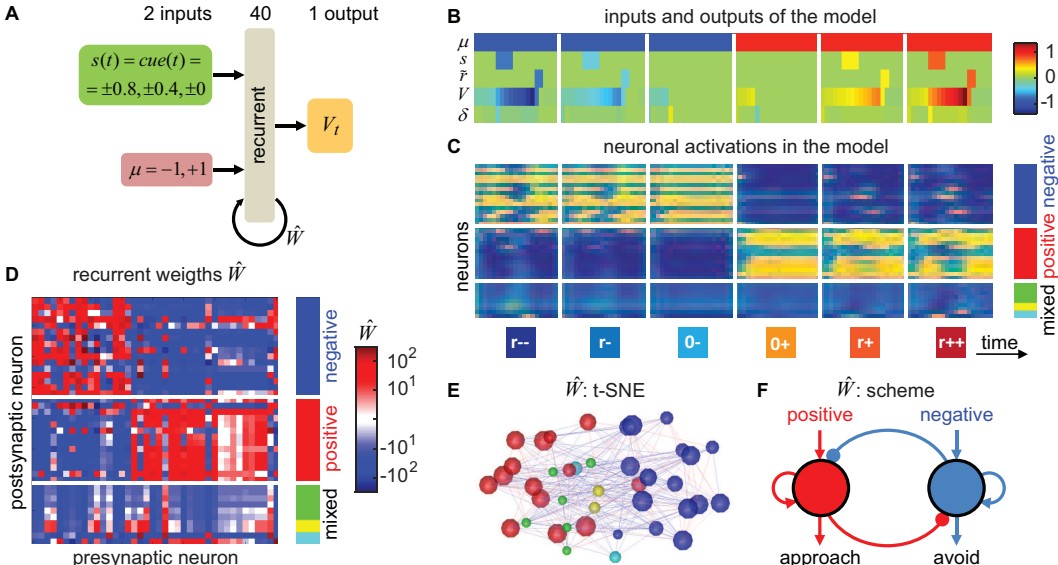

Figure 7: (A) The architecture of the RNN computing the Q-function in the Pavlovian conditioning task. (B) Inputs and outputs of the RNN for each trial type. Inputs: motivation, cue, reward. Outputs: sample V-function. Bottom row: TD error. Trial types (left to right): large negative reward, small negative reward, no negative reward, no positive reward, small positive reward, large positive reward. (C) Sample responses of neurons in the RNN clustered into the negative motivation (blue), positive motivation (red) and neurons of mixed sensitivity (other). (D) Recurrent connectivity matrix. (E) t-SNE representation. (F) Push-pull circuit.

## 3 DISCUSSION

Motivation has been defined previously as the need-based modulation of reward magnitude. Here we propose an RL approach to the neural networks that can be trained to include motivation into the calculation of action. We consider a diverse set of example networks that can solve different problems following a similar pattern. We train such networks using TD rule via conventional backpropagation. We find that the networks can learn optimal behaviors, including behaviors that reflect complex scenarios of future motivation changes. When compared to the responses of neurons in the mouse brain, our neural network model can accurately predict behavioral outcomes, demonstrates similar patterns of neuronal responses, and generates predictions for network connectivity.

We trained our networks to compute future motivation-dependent reward in the Pavlovian conditioning task. Connecting RL – and, in particular, TD methods – to Pavlovian conditioning tasks was a matter of the extensive research, reviewed by Sutton & Barto (1987). We found that the neurons in the RNNs trained to recognize motivation can be clustered into 2 oppositely tuned populations: neurons increasingly active in positive and negative reward trials. In agreement with this finding, we found similar two groups of neurons in the mouse VP: a basal ganglia region implicated in motivation-dependent estimates of reward (Richard et al., 2016). Thus, neural networks with motivation, trained to perform in realistic tasks, develop responses similar to those in the brain.

The recurrent network structure in this Pavlovian conditioning case is compatible with the conventional models of working memory. The information about upcoming reward – once supplied by a cue – is maintained in the network due to the positive recurrent feedback. This feedback is produced by inhibition between two oppositely tuned populations of neurons, i.e. positive and negative reward sensitive cells. Thus, the experimentally observed presence of particular neural populations may be

a consequence of the functional requirements on the network to maintain persistent variables within a trial. This function is reflected in both neural responses and architecture. Our findings present a generative hypothesis for how information about trial outcome is maintaned in the brain networks.

In recent work, Keramati & Gutkin (2014) show that *homeostatic RL* explains prominent motivation-related behavioral phenomena including anticipatory responding (Mansfield & Cunningham, 1980), dose-dependent reinforcement and potentiating effect of deprivation (Hodos, 1961), inhibitory effect of irrelevant drives (Dickinson & Balleine, 2002), etc. Although homeostatic RL defines the rewards as the gradients of the cost function with a fixed point, the theoretical predictions generalize to the models with linear, or approximately linear, multiplicative motivation. We therefore expect the behaviors of our models to be consistent with the large body of experimental data mentioned above.

Motivation offers a framework compatible with other methods in machine learning, such as R-learning, goal-conditioned RL, and hierarchical RL (HRL). In *R-learning*, (Sinakevitch et al., 2018; Schwartz, 1993), the cumulative sum of future rewards is computed with respect to the average level. The average reward level is a slowly changing variable computed across several trials, which makes it similar to motivation. In *goal-conditioned RL* – the closest counterpart to RL with motivation – the Q-function depends on three parameters: $Q(\vec{s}_t, a_t, g)$, where $g$ is the current static goal. In the motivation framework, multiple dynamic goals are present at the same time, and it is up to an agent to decide which one to pursue. *HRL methods* include the options framework (Sutton & Barto, 1998; Sutton et al., 1999), RL with subgoals (Sutton et al., 1999), feudal RL (Dayan & Hinton, 2000; Bacon & Precup, 2018), and others. In HRL, complex tasks are solved by breaking them into smaller, more manageable pieces. HRL approaches have several advantages compared to traditional RL, such as transfer of knowledge from already learned tasks and the ability to faster learn solutions to complex tasks. Although HRL methods are computationally efficient and generate behaviors separated into multiple levels of organization – which resemble animals' behavior – a mapping of HRL methods to brain networks is missing. Here, we suggest that motivation offers a way for HRL algorithms to be implemented in the brain. In case of motivation, both manager and lower-level actor nerworks receive the same reward, which makes motivated networks different from e.g. their feudal counterparts (Dayan & Hinton, 2000; Bacon & Precup, 2018).

As described above, actions in the motivation-based RL are selected on the basis of Q-function $Q(s_t, a, \mu)$. An action $a_t$ selected at certain time maximizes the Q-function, representing the total expected future reward, and leads to the transition of the agent to the new state: $s_t \rightarrow a_t \rightarrow s_{t+1}$. Because of the dependence of the Q-function on motivation, the action choice depends on the variable $\mu$ representing motivation in our framework. We argued above that motivation allows RL to have the flexibility of a rapid change in behavioral policy when the need of an animal fluctuates. The same mechanism can be used to implement HRL, if motivation $\mu$ is supplied by another, higher-level "manager" network with its own Q-function, $Q^{(1)}(\mu_t, a^{(1)}, \mu^{(1)})$. When the higher-level network picks an action $a_t^{(1)}$, it leads to a change in the motivational state for the lower-level network: $\mu_t \rightarrow a_t^{(1)} \rightarrow \mu_{t+1}$ thus rapidly changing the behavior of the latter. The "manager" network could on its own be controlled by a higher-level manager via its own motivation $\mu^{(1)}$. Such decision hierarchy may include several management levels, with the dynamics of motivation on level $l$ determined via Q-function computed on level $l + 1$: $Q^{(l+1)}(\mu_t^{(l)}, a^{(l+1)}, \mu^{(l+1)})$ and $\mu_t^{(l)} \rightarrow a_t^{(l+1)} \rightarrow \mu_{t+1}^{(l)}$. Although HRL is outside the scope of this project, we suggest that motivation-based RL studied here may link the neurobiology of adaptive behaviors to developments in machine learning.

Overall, we suggest that motivation-based networks may generate complex ongoing behaviors that can adapt to dynamic changes in an organism's demands. Thus, neural networks with motivation can both encompass more complex behaviors than networks with a fixed reward function and be mapped onto animals's circuits that control rewarded behaviors. Since animal performance in realistic conditions depends on the states of satiety, wakefulness, etc., our approach should help build more realistic computational models that include these variables. Importantly, when we compared the responses of neurons in the mouse brain to our model, our neural network model can accurately predict behavioral outcomes, demonstrates similar patterns of neuronal responses, and generates predictions for network connectivity. In particular, our model explains why basal ganglia neurons form two classes: tuned to positive and negative rewards. In our model, these classes emerge from the need to maintain the information about future beward within the trial using positive recurrent feedback. Thus, networks with motivation considered here give imporant insights into the mechanisms of signal processing in brain reward circuits.

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

# A    APPENDIX – METHODS

## A.1    THE FOUR DEMANDS TASK

To optimize the behaviors in the Four Demands task, we implemented a feedforward neural network as described below. On the input, the network received an agent's state and motivation. The state variable contained an agent's position, which was represented by a 36-dimensional one-hot vector. The motivation was represented by a 4-dimensional integer vector. From both state and motivation variables, we subtracted the mean values. To balance the contributions of state and motivation to the network, we normalized their variances to 1 and 9 respectively, since the ratio of the number of these variables is 4/36 (in case of non-motivated agents, we set the motivation variable to zero). The inputs of the network were propagated through three hidden layers (100 sigmoid units each), and an output layer (5 linear units). We trained the network to compute the Q-values of the possible actions: to move left, right, up, down, or to stay.

On every iteration, we picked an action, corresponding to the largest network output (Q-value). With probability $\varepsilon$, we replaced the selected action with a random action ($\varepsilon$-greedy policy; $\varepsilon$ decreased exponentially from 0.5 to 0.05 throughout simulation; in case of random walk agents, we set $\varepsilon = 1$). If the selected action resulted in a step through a "wall", the position remained unchanged; otherwise we updated the agent's position. For the agent's new position, we computed the perceived reward $(\vec{r} \cdot \vec{\mu}^T)$, and used Bellman equation ($\gamma = 0.9$) to compute TD error. We then backpropagated the TD error through the network to update its weights (initialized using Xavier rule). We performed $4 \cdot 10^5$ training iterations with the learning rate decreasing exponentially from $3 \cdot 10^{-3}$ to $3 \cdot 10^{-5}$.

We trained the network using various motivation schedules as follows. Each component of the motivation was increased by one on every iteration. If a component of motivation $\mu_n$ reached the threshold $\theta_n$, we stopped increasing this component any further. If the reward of a type $n$ was consumed on current iteration, we dropped the corresponding component of motivation $\mu_n$ to zero. For motivated, non-motivated, and random walk agents, we trained 41 model each (123 models total) with motivation thresholds $\theta_1 = \theta_2 = \theta_3 = \theta_4$ ranging from 1 to 100, spaced exponentially, one training run per unique $\theta$ value. To mimic addiction, we also trained a model with $\theta_1 = \theta_2 = \theta_3 = 1$, and $\theta_4 = 10$. For each run, we displayed sequences of agent's locations to establish correspondence between policies and average reward rates.

## A.2    THE TRANSPORT NETWORK TASK

To build an environment for the transport network task, we defined the locations for 10 "cities" by sampling $x$ and $y$ coordinates from the standard normal distribution $N(0, 1)$. For these locations, we computed Delaunay triangulation to define a network of the roads between the cities. For each road (Delaunay graph edge), we computed its length – the Euclidean distance between two cities it connects. We then selected multiple random subsets of 3 cities to be visited by an agent: the training set ($10^4$ target subsets), and the testing set (50 different target subsets).

To navigate the transport network, we implemented a feedforward neural network as described below. On the input, the network received an agent's state and motivation. The state variable contained an agent's position, which was represented by a 10-dimensional one-hot vector. The motivation was represented by a 10-dimensional binary vector. To specify the agent's targets, we initialized the motivation vector with 3 non-zero components $\mu_{i_1}...\mu_{i_3}$, corresponding to the target cities $i_1...i_3$. The inputs of the network were propagated through a hidden layer (200 Leaky ReLU units; leak $\alpha = 0.01$), and an output layer (10 linear units). We trained the network to compute the Q-values of the potential actions (visiting each of the cities).

On every iteration within a task episode, we picked an action to go from the current city to one of the immediately connected cities, then we updated the current position. To choose the action, we used the softmax policy ($\beta = 0.5$) over the Q-values of the available moves. When the motivation $\mu_j$ towards the new position $j$ was non-zero, we yielded the reward of 5, and dropped the motivation $\mu_j$ to 0. On every iteration, we reduced the reward by the distance travelled within this iteration. The task episode terminated when all the components of motivation were equal to zero. On every iteration, we used Bellman equation ($\gamma = 0.9$) to compute the TD error. We backpropagated the TD error through the network to update its weights (initialized using Xavier rule). Overall, we performed training on $10^4$ task episodes with the learning rate $10^{-2}$. To assess the model performance, we evaluated the model on the testing set and compared the resulting path lengths to the precomputed shortest path solutions.

## A.3    PAVLOVIAN CONDITIONING TASK

To build a circuit model of motivation in Pavlovian conditioning task, we implemented a recurrent neural network. We trained the network on terminating sequences of 20 iterations, representing time within individual trials. On the input, the network received an agent's state and motivation. The state variable contained a cue (conditioned stimulus; CS), which we chose randomly from $\{\pm 0.0, \pm 0.4, \pm 0.8\}$. Depending on iteration, the state variable was equal either to the CS (iterations 6-9 out of 20), or to zero (elsewhere). The motivation variable $\mu = \pm 1$ was equal to the sign of the CS; it was constant throughout the entire sequence of 20 iterations. The inputs of the network were propagated through a recurrent layer (40 sigmoid units), and an output layer (1 linear unit). We trained the network to compute the V-values for each iteration within the sequence.

On every iteration, we computed a reward, reflecting the reward (unconditioned stimulus; US). Depending on iteration, the reward was equal either to the CS (iterations 15-16), or to zero (elsewhere). We used the rewards in Bellman equation $\gamma = 0.9$ to compute a TD error for every iteration. We then backpropagated the TD errors through time to update the network's weights (initially drawn from the uniform distribution $U(-10^{-5}, 10^{-5})$). We performed training on $3 \cdot 10^5$ minibatches of 20 sequences each, with the learning rate of $10^{-1}$.

We then clustered the recurrent neurons after training as follows. First, for every neuron we computed 6 average activations, corresponding to the unique types of trials (positive/negative motivation with zero/small/large reward). Then, we used the average activations to compute a correlation matrix for the neurons. Finally, we processed the correlation matrix with the watershed algorithm (marker-based; $h = 0.04$), hence clustered the recurrent neurons. To examine the connectivity between the clusters, we used the weights of the recurrent neurons to compute a new correlation matrix. We then applied t-SNE in 3 dimensions ($p = 30$), and color-coded the neurons with respect to the clusters.

