# OpenReview forum: "Neural networks with motivation"
_ICLR.cc/2020/Conference — Reject_

### Official Review · AnonReviewer1 · 2019-10-18
**Official Blind Review #1**

**Rating:** 3

**Review:**

This paper builds a model of motivation-dependent learning.  A motivation channel is provided as an additional input to and RL-based learning system (essentially concatenated to state information), similar to goal-conditioned approaches (as the authors mention).  The motivational variables evolve according to their own rules, and are designed/interpreted as biological motivations such as water, food, sleep and work.  While the narrative is interesting, I lean towards reject as I believe it failed to deliver on what it promised.

In the first experiment, the satisfaction of these motivations are mapped onto a 4-room setting, where being in each room satisfies a motivation.  The choice to map the four rooms to biological drives is cute, but possibly confusing/misleading since this navigation problem really has nothing to do with these biological drives. A claim is that by providing the motivation as input to the policy, it is more robustly (across seeds) able to learn the "migration" (i.e. cycling) behavior among the rooms.  In a second example, a similar problem is solved involving navigation on a graph.

The final, most substantial example, is a policy trained to solve a simple, abstract version of a behavioral task. In this setting, a motivation channel was again used.  However, the motivation channel value is now fixed to one of two discrete values, essentially meaning it is simply a task-label variable, a paradigm that has already been applied in the context of simple models of neuroscience tasks, e.g. see Song et al. 2017 "Reward-based training of recurrent neural networks for cognitive and value-based tasks".

There is a bit of a mixed framing overall as to whether it is being claimed that the "motivation" being passed as an input is a fundamental contribution to AI/RL (I think it is not), versus the computational modeling of biological motivation.  I think the people qualified to judge whether the computational model is a worthwhile model of motivation specifically are probably a narrower set of computational neuroscientists.  I do think there is value in the kind of computational modeling performed, involving establishing a relationship between training a neural network to solve a behavioral task and comparing this with real neural data.  This paradigm already becoming increasingly popular within computational neuroscience.  However, while I find the results slightly interesting, but not very significant, as someone interested in the biology of motivation, I question whether the nature of these contributions would be of broad interest at this venue.

More fundamentally, I don't believe there is a meaningful ML/AI/RL contribution, and I have some issues with the presentation of the first two examples.  While I do like the narrative inspiring these problems, I find the implementations of the problems too simplified to really be meaningfully related to their inspiration (in terms of motivated behaviors).  Rather than really model motivation as part of the policy architecture, the authors have proposed a solution to modeling motivation that makes motivation a feature of the environment.  Essentially, the reward provided by the environment depends on an extra latent variable and by hiding this (in the cases where the policy does not see motivation inputs), it is quite likely that it becomes too difficult for the value function to predict what is happening (the environment has become partially observed).  This seems less a setting where motivation channels solve a problem, and more just an example of an environment that has more complex rules for generating rewards being more challenging to learn about, especially if latent variables are not available to the value function.  Critically, it has not been shown that motivational systems are useful for artificial agents, rather the tasks themselves have been designed to attempt to be models of biological motivation.

Personally, I am interested in motivated behaviors and think that future AI developments should take note of this field, but again, the present work does not provide actionable insights into implementing an artificial motivation system.  At the same time, this work does not provide interesting enough neurobiological results for those to stand on their own either.

Minor clarification:

"trained to perform in realistic tasks" -- the task is very simple.  I would consider this a fairly abstract model of the task.


**Experience Assessment:**

I have published in this field for several years.

**Review Assessment: Checking Correctness Of Derivations And Theory:**

N/A

**Review Assessment: Checking Correctness Of Experiments:**

I assessed the sensibility of the experiments.

**Review Assessment: Thoroughness In Paper Reading:**

I read the paper at least twice and used my best judgement in assessing the paper.

---

> ### Author Response · Authors · 2019-11-11
> **Official Response to the Blind Review #1**
>
> We thank the reviewer for careful reading of our paper at least twice and providing the thorough, meaningful, and in depth review. We would like to debate, however, the assessment of value of this work for both neuroscience and machine learning communities and its relevance to the venue. We agree with the reviewer that “future AI developments should take a note” of motivated behaviors. Our overall goal is to facilitate this exchange. It is also clear that neuroscience community should take notice of many developments in AI. One of such developments in the hierarchical RL (HRL) that has not been mapped on neural circuits yet. Our paper proposes that motivational salience, which we call ‘motivation’ for brevity, may have evolved from modulating simple feeding behaviors to solving more complex hierarchical cognitive tasks. As such, our paper aims at bridging the gap between HRL community in ML and neuro communities interested in understanding motivated behaviors. Clearly, building a motivated HRL model is not a task for a single paper. Our goal was therefore to introduce the concept of motivational salience to the ML community. In addition, we believe, we have made substantial contributions to computational neuroscience and to understanding of computational algorithms run by circuits in ventral pallidum (VP) as detailed below
>
> 1) Our paper presents the first example of neural network processing information about motivational salience. Motivational salience has been described in RL framing before, but the networks processing motivation are missing. The reviewer may not agree with how we frame the problem, but, perhaps, it is also important to formulate one solution in order to compel community to find alternatives. Our solution is useful, however, because it helps solve complex computational tasks and allows to make sense of responses of neurons in basal ganglia.
>
> 2) We explain the presence of two oppositely-tuned populations of neurons in VP as resulting from the need to solve temporal credit assignment problem via maintaining working memory about reward expectation between CS and US.
>
> 3) Our general framework allows to derive clear experimentally testable predictions about network structure in VP. We use a conventional machine learning algorithm (recurrent network training using backpropagation) to derive the structure of the network in VP from the first principles and show that it should contain inhibitory connections between two populations of neurons. We agree with the reviewer that backpropagation has been used to train recurrent RL V-networks before, for example, in the neuroscience setting by Song et al. (2017). However, Song et al. (2017) did not show that the network has a push-pull architecture. Since this particular architecture is known to be important in neural systems, it is valuable to show that the same connectivity can be used in circuits implementing motivated behavior. We argue that the presence of the push-pull circuit leads to the emergence of the two oppositely tuned populations of neurons.
>
>
> Overall we suggest that our paper introduces motivational salience as a potential basis of HRL, uses the general machine learning framework based on motivational salience to explain existing experimental data (two populations of neurons), and generates clear experimentally testable predictions about the structure of network of real biological neurons in basal ganglia. We thus humbly suggest that it makes a substantial contribution to the understanding of the circuit basis of computation involved in motivated behaviors.

---

> > ### Comment · AnonReviewer1 · 2019-11-12
> > **Reply to response.**
> >
> > I acknowledge the author response.  I do think the neural data comparison is meaningful, but given my other concerns with the proposed framework, the initial experiments, and the issues concerning the relationship between the first and second parts of the paper, I stand by my initial review and score.

---

> > > ### Author Response · Authors · 2019-11-12
> > > **Re: reply to response.**
> > >
> > > We highly appreciate the reviewer’s comments, and the prompt reply. We hope to have convinced the reviewer that our paper makes meaningful, if not significant, contributions to understanding the circuit basis of motivated behaviors. First, we formulate – for the first time – a network model that processes the information regarding the motivational salience. Second, using this model, we explain two oppositely tuned populations of the cells in the ventral pallidum (VP), which we observed experimentally. We argue that the presence of these two populations helps solving the temporal credit assignment problem by maintaining persistent activity in the VP network. This argument has never been presented before. Finally, we make the experimentally testable predictions about the VP circuits containing these two populations of cells. Thus, our theory can be verified in the experiment. Our arguments are derived from the first principles and connect the neurobiological observations with a common ML framework (deep learning), and a novel RL approach (motivational salience). Since establishing such connections is an important part of the ICLR meeting agenda, we argue that our paper is a perfect fit for this venue.
> > >
> > > Our framework allows us to build the networks featuring the addiction-related behaviors (Figure 3). Machine learning models of the networks with addiction have never been presented before. Despite simplicity, our models allow generating a variety of complex behaviors, including migration, binging, addiction, optimal path finding, maintaining the working memory, etc.
> > >
> > > The reviewer is concerned that we interpret motivation as a part of the environment: “Rather than really model motivation as part of the policy architecture, the authors have proposed a solution to modeling motivation that makes motivation a feature of the environment.” We agree with this summary; however, for the first step, we aimed modeling motivation in a setting close to the biological reality. Motivational salience – e.g. in simple feeding/foraging tasks – reflects the inputs from outside the (VP) network. As such, these inputs are indistinguishable from the environment for the network. For example, the sense of thirst – being the main driving motivation in our experiments – aggregates in the median preoptic nucleus (MnPO) of hypothalamus from various thirst neurons, responding to the changes in homeostasis (blood pressure/composition/osmosis, etc.) As such, the thirst-related component of motivational salience is external w.r.t. the VP – and is better represented by the external (“environmental”) variables in our models. We understand the reviewer’s interest in hierarchical models for motivational salience – to include motivation as a part of the policy architecture. Such models, however, are outside the scope of this paper, which attempts to describe the first logical step in modeling motivation.
> > >
> > > Similarly, in our experimental work, we rely on classical conditioning (CC), which is a traditional, albeit simple, way of studying RL in biological setting. Despite the simplicity, our experiments provide rich data on neuronal responses with respect to both reward and motivational salience. Whether the experimental paradigm is interesting or not is a subjective matter. We believe that an experimental paradigm is meaningful as long as it allows testing models and comparing their results to experimental data. The data that we collected in the CC paradigm does allow us to compare the results of motivation-based models to the neuronal responses; it also allows us to derive predictions. A simple experimental paradigm, as the CC, is a necessary first step in modeling motivation; without implementing it, any further studies would be premature.
> > >
> > > The reviewer also comments on the relationship between the first and the second parts of the paper, which, they are concerned, is lacking. We would like to point out that there is a strong connection between the pure RL part and the CC experiments. Mathematically, both parts use the same method, based on motivational salience. The only difference is the use of the recurrent networks in the CC part – to reflect the VP neurons forming the recurrent connections. Thus, two parts of the paper are linked by using the same novel ML paradigm.
> > >
> > > Considering the arguments above, the authors would like to humbly request the reviewer to reconsider their valuation.

---

### Official Review · AnonReviewer2 · 2019-10-22
**Official Blind Review #2**

**Rating:** 3

**Review:**

The authors investigate mechanisms underlying action selection in artificial agents and mice. To achieve this goal, they use RL to train neural networks to choose actions that maximize their temporally discounted sum of future rewards. Importantly, these rewards depend on a motivation factor that is itself a function of time and action; this motivation factor is the key difference between the authors' approach and "vanilla" RL. In simple tasks, the RL agent learns effective strategies (i.e., migrating between rooms in Fig. 1, and minimizing path lengths for the vehicle routing problem in Fig. 5).

The authors then apply their model to a task in which the agent is presented with sound cues. Depending on the trial block, the reward for the given cue is either zero, positive, or negative; the authors suggest that these varying reward values correspond to varying motivational states. In this setting, the model learns to have two populations of units; each selective to either positive or negative rewards. Recurrent excitation within populations and mutual inhibition between populations define the learned dynamics.

Finally, the authors train mice on this same task, and record from neurons in area VP. Those neurons show a similar structure to the RNN: subpopulations of neurons respond to either positive or negative rewards.

First, I'd like to thank the authors for the excellent clarity of this paper. It was very clear, and interesting to read.

I have some suggestions for how to deepen the connection between the model and the experiment, and some concerns about the necessity of the motivation framework to the Pavlovian task:

1) The authors make the prediction that neurons in VP should show (functional) connectivity matching that learned by their model. This could be tested in their data. If that prediction is true, then one should see positive noise correlations for neuron pairs of the same preference (i.e., within the same pool, defined by spiking more for positive, or for negative rewards), and negative noise correlations for pairs of neurons with different preferences (i.e., one neuron in each pool).

2) A recent preprint by Sederberg and Nemenman (doi: https://doi.org/10.1101/779223) argued against over-interpreting the stimulus selectivity of neurons in recurrent circuits. They showed that, even in randomly connected (untrained) networks: a) neurons show either positive or negative selectivity; and b) neuron pairs with selectivity for the same stimulus (or task) feature tend to excite each other, and neuron pairs with opposite selectivity tend to inhibit each other.  Given that finding, I wonder how compelling is the match between the mouse data and the RL agent (Figs. 6 and 7): could randomly-connected untrained networks show similar phenomena as in the mouse (Fig. 6)?

I'm not asking if the untrained network can duplicate all the details of the trained one in Fig. 7. Just whether the mouse data could be recapitulated by a simpler (no training) model.

3) For the Pavlovian conditioning in the RL agent, I'm not sure I'd describe this as changing motivation. It seems instead that the (external) reward contingency really changes between states. So the fact that the same network can make predictions in both cases seems more like metalearning than motivation-based action selection. For this reason, it's hard for me to connect the two halves of the paper: the first half has nice ideas on motivation-based action selection, while the second one has no apparent action selection, and hence no mechanism for the agent's motivation to matter.


**Experience Assessment:**

I have published one or two papers in this area.

**Review Assessment: Checking Correctness Of Derivations And Theory:**

N/A

**Review Assessment: Checking Correctness Of Experiments:**

I assessed the sensibility of the experiments.

**Review Assessment: Thoroughness In Paper Reading:**

I read the paper thoroughly.

---

> ### Author Response · Authors · 2019-11-11
> **Official Response to the Blind Review #2 - part 1**
>
> The authors would like to sincerely thank the Reviewer #2 for their time, interest in the paper, and thorough comments. Below we attempted to address all of the reviewer specific concerns.
>
> 1) "The authors make the prediction that neurons in VP should show (functional) connectivity matching that learned by their model. This could be tested in their data. If that prediction is true, then one should see positive noise correlations for neuron pairs of the same preference (i.e., within the same pool, defined by spiking more for positive, or for negative rewards), and negative noise correlations for pairs of neurons with different preferences (i.e., one neuron in each pool)."
>
> The authors agree with the rationale for the proposed analysis. Unfortunately, the full data containing the individual spikes is not yet available for analysis due to handling delays; instead, we operated on the average activations for each cell (Fig. 6C, D, F). To this end, in our original submission we did a simpler version of what the Reviewer #2 proposed – though we did not expand its description due to the space limitations. Namely, we averaged activity of each cell under each experimental condition, and then used correlations of the average activities to cluster cells in the data. This way, activities of the cells within a same cluster had positive correlation, as anticipated by the Reviewer #2; the property of being positive/negative with respect to reward was established through visually evaluating cell activations in each cluster (Fig. 6C, D, F).
> To include an explicit description of correlation-based clustering procedure in the paper, we included the following statement in the newly written Methods appendix: “We then clustered the recurrent neurons after training as follows. First, for every neuron we computed 6 average activations, corresponding to the unique types of trials (positive/negative motivation with zero/small/large reward). Then, we used the average activations to compute a correlation matrix for the neurons. Finally, we processed the correlation matrix with the watershed algorithm (marker-based; h = 0.04), hence clustered the recurrent neurons”.
>
> 2) "A recent preprint by Sederberg and Nemenman (doi: https://doi.org/10.1101/779223) argued against over-interpreting the stimulus selectivity of neurons in recurrent circuits. They showed that, even in randomly connected (untrained) networks: a) neurons show either positive or negative selectivity; and b) neuron pairs with selectivity for the same stimulus (or task) feature tend to excite each other, and neuron pairs with opposite selectivity tend to inhibit each other.  Given that finding, I wonder how compelling is the match between the mouse data and the RL agent (Figs. 6 and 7): could randomly-connected untrained networks show similar phenomena as in the mouse (Fig. 6)? I'm not asking if the untrained network can duplicate all the details of the trained one in Fig. 7. Just whether the mouse data could be recapitulated by a simpler (no training) model."
>
> We appreciate the Reviewer #2 pointing out the paper by Sederberg and Nemenman, highlighting the issue of interpreting stimulus selectivity in recurrent circuits. Although we could not account for this work in our original submission – as it was published two days before the conference deadline – the authors agree that it is now reasonable to perform additional analysis, as suggested by the Reviewer #2.
> In agreement with the conclusions of the paper, when we substitute the trained weights in the model with the random weights, we observe positive- and negative-selectivity cells. In particular, every cell has two tunings – to motivation, and to cue (same as reward) – reflecting two separate inputs to the neural network. In random (no training) models, these inputs are propagated through independent random weights of both signs. As a result – and we observe it in simulation – selectivity to motivation and selectivity to reward are independent, e.g. positive-motivation cell may be at the same time negatively tuned to the reward. Therefore, in total, random connectivity in our task yields four clusters of cells: (positively/negatively tuned to motivation/reward). On contrast, in the data, we only observe two clusters: positive-motivation cells are always positively tuned to reward, and negative-motivation cells are always negatively tuned to reward. Thus, a non-trained model does not recapitulate the mouse data.

---

> > ### Author Response · Authors · 2019-11-11
> > **Official Response to the Blind Review #2 - part 2**
> >
> > 3) "For the Pavlovian conditioning in the RL agent, I'm not sure I'd describe this as changing motivation. It seems instead that the (external) reward contingency really changes between states."
> >
> > In our version of Pavlovian conditioning task, reward contingency and motivation are highly correlated by design. The data, however, cannot be explained strictly in terms of the reward. Motivation is an essential part of the model that cannot be omitted. Specifically, in both positive-motivation and negative-motivation blocks of trials, there is a group of trials in which the animal does not receive reward or punishment. The only variable that distinguishes these two groups of trials (no reward / no punishment) is motivation. Despite the same trial outcome in these two cases (the same reward contingency), neuronal and behavioral readouts are different. Firing rates in the VP neurons differ significantly both before and after the cue (Fig. 6D, F, lines “0+” and “0-“). The animals’ behaviors in zero-reward and zero-punishment trials differ too: in negative-motivation blocks of trial, the animals show eye-blinking response (although there is no air puff), whereas in positive-motivation blocks of trials, the animals show reward tube licking response (although there is no water reward; data not shown). Our model also learns to differentiate these two cases (Figure 7B). Thus, in addition to the changed reward contingency, motivation is a necessary variable for our Pavlovian conditioning task, reflected in behavior and neuronal activity.
> >
> > "So the fact that the same network can make predictions in both cases seems more like metalearning than motivation-based action selection. For this reason, it's hard for me to connect the two halves of the paper: the first half has nice ideas on motivation-based action selection, while the second one has no apparent action selection, and hence no mechanism for the agent's motivation to matter."
> >
> > The authors appreciate the argument presented above. In fact, we view the ability to generate complex ongoing behaviors based on simple conditions – and to generalize those across similar conditions – as a major advantage of the Motivation framework, resulting from separation of fast-changing variables (state), and slow-changing variables (motivation). As the Reviewer #2 correctly points out, in classical conditioning case, animals definitely cannot take action to disrupt the sequence of the rewards. What they can do instead, is to prepare for the upcoming positive/negative reward, and take action to exploit/mitigate its effect (reward tube licking / eye blinking; data not shown due to space constraints). We do however have a much better readout of biological network’s behavior, i.e. neuronal responses. We show that the model network with motivation can explain responses of neurons in the real biological network in the conditions where motivation is a non-redundant behavioral variable. We therefore believe that including motivated networks with classical conditioning task is relevant to our paper.  We are presently working on setting up motivated behavioral tasks in which the animal can actually act (besides licking and blinking). We believe however, that CS task is an important milestone that should be studied in the framework of motivational salience before more complex tasks are considered.

---

### Official Review · AnonReviewer3 · 2019-11-06
**Official Blind Review #3**

**Rating:** 1

**Review:**

This paper presents a computational model of motivation for Q learning and relates it to biological models of motivation. Motivation is presented to the agent as a component of its inputs, and is encoded in a vectorised reward function where each component of the reward is weighted. This approach is explored in three domains: a modified four-room domain where each room represents a different reward in the reward vector, a route planning problem, and a pavlovian conditioning example where neuronal activations are compared to mice undergoing a similar conditioning.

Review Summary:
I am uncertain of the neuroscientific contributions of this paper. From a machine learning perspective, this paper has insufficient details to assess both the experimental contributions and proposed formulation of motivation. It is unclear from the discussion of biological forms of motivation, and from the experimental elaboration of these ideas, that the proposed model of motivation is a novel contribution. For these reasons, I suggest a reject.

The Four Rooms Experiment:

In the four-rooms problem, the agent is provided with a one-hot encoding representing which cell it the agent is located in within the grid-world. The reward given to the agent is a combination of the reward signal from the environment (a one-hot vector where the activation is dependent on the room occupied by the agent) and the motivation vector, which is a weighting of the rooms. One agent is given access to the weighting vector mu in its state vector: the motivation is concatenated to the position, encoding the weighting of the rooms at any given time-step. The non-motivated agent does not have access to mu in its state, although its reward is weighted as the motivated agent’s is. The issue with this example is that the non-motivated agent does not have access to the information required to learn a value-function suitable to solve this problem. By not giving the motivation vector to non-motivated agent, the problem has become a partially observable problem, and the comparison is now between a partially observable and fully observable setting, rather than a commentary on the difference between learning with and without motivation.

In places, the claims made go beyond the results presented. How do we know that the non-motivated network is engaging in a "non-motivated delay binge"? We certainly can see that the agent acquires an average reward of 1, but it is not evident from this detail alone that the agent is engaging in the behaviour that the paper claims.

Moreover, the network was trained 41 times for different values of the motivation parameter theta. Counting out the points in figure 2, it would suggest that the sweep was over 41 values of theta, which leaves me wondering if the results represent a single independent trial, or whether the results are averaged over multiple trials. Looking at the top-right hand corner I see a single yellow dot (non-motivated agent) presented in line with blue (motivated agent) suggesting that the point is possibly an outlier. Given this outlier, I’m led believe that the graph represents a single independent trial. A single trial is insufficient to draw conclusions about the behaviour of an agent.

The Path Routing Experiment:

In the second experiment, where a population of agents is presented in fig 5, it is claimed that on 82% of the trials, the agent was able to find the shortest path. Looking at the figure itself, at the final depicted iteration, all of the points are presented in a different colour and labelled “shortest path”. The graph suggests that 100% of the agents found the shortest path. The claim is made that for the remaining 18% of the agents, the agents found close to the shortest path—a point not evident in the figures presented.


Pavlovian Conditioning Experiment:

In the third experiment, shouldn’t Q(s) be V(s)? In this setting, the agent is not learning the value of a state action pair, but rather the value of a state. Moreover, the value is described as Q(t), where t is the time-step in the trial; however, elsewhere in the text it is mentioned that the state is not simply t, but contains also the motivation value mu.

The third experiment does not have enough detail to interpret the results. It is unclear how many trials there were for both of the prediction settings. It is unclear whether the problem described is a continuing problem or a terminating prediction problem—i.e., whether after the conditioned stimulus and unconditioned stimulus are presented to the agent, does the time-step (and thus the state) reset to 0, or does time continue incrementing? If it is a terminating prediction problem, it is unclear whether the conditioned stimulus and unconditioned stimulus were delivered on the same time-steps for each independent trial. If I am interpreting the state-construction correctly, the state is incrementing by one on each time-step; this problem is effectively a Markov Reward Process where the agent transitions from one state to the next until time stops with no ability to transition to previous states.

In both the terminating and continuing cases, the choice of inputs is unusual. What was the motivation for using the time-step as part of the state construction?

How is the conditioned stimulus formulated in this setting? It is mentioned that it is a function of time, but there are no additional details.

From reading the text, it is unclear whether fig 7b/c presents activations over multiple independent trials or a single trial.

General Thoughts on Framing:

This paper introduces non-standard terms without defining them first. For example, TD error is introduced as Reward Prediction Error, or RPE: a term that is not typically used in the Reinforcement Learning literature. To my understanding, there is a hypothesis about RPE in the brain in the cognitive science community; however, the connection between this idea in the cognitive science literature and its relation to RL methods is not immediately clear.

Temporal Difference learning is incorrectly referred to as "Time Difference" learning (pg 2).

Notes on technical details:

- The discounting function gamma should be 0<= gamma <=1, rather than just <=1.

- discounting not only prevents the sum of future rewards from diverging, but also plays an important role in determining the behaviour of an agent---i.e., the preference for short-term versus long-term rewards.

- pg 2 "the motivation is a slowly changing variable, that is not affected substantially by an average action" -- it is not clear from the context what an average action is.

- Why is the reward r(s|a), as opposed to r(s,a)?

Notes on paper structure:

- There are some odd choices in the structure of this paper. For instance, the second section---before the mathematical framing of the paper has been presented---is the results section.

- In some sentences, citations are added where no claim is being made; it is not clear what the relevance of the citation is, or what the citation is supporting. E.g., “We chose to use a recurrent neural network (RNN) as a basis for our model” following with a citation for Sutton & Barto, 1987.

- In some sentences, citations are not added where substantial claims are being made. E.g, “The recurrent network structure in this Pavlovian conditioning is compatible with the conventional models of working memory”. This claim is made, but it is never made clear what the conventional computational models of working memory are, or how they fit into the computational approaches proposed.

- Unfortunately, a number of readers in the machine learning community might be unfamiliar with pavlovian conditioning and classical conditioning. Taking the time to unpack these ideas and contextualise them for the audience might help readers understand the paper and its relevance.

- Figure 7B may benefit from displaying not just the predicted values V(s), but a plot of the prediction over time in comparison to the true expected return.


**Experience Assessment:**

I have published one or two papers in this area.

**Review Assessment: Checking Correctness Of Derivations And Theory:**

N/A

**Review Assessment: Checking Correctness Of Experiments:**

I carefully checked the experiments.

**Review Assessment: Thoroughness In Paper Reading:**

I read the paper thoroughly.

---

> ### Author Response · Authors · 2019-11-11
> **Official Response to the Blind Review #3 - part 1**
>
> The authors would like to thank the Reviewer #3 for their time, and attention to the details. We believe that the Reviewer #3’s comments helped make the manuscript more rigorous.
>
> Please find the specific responses below.
>
> “I am uncertain of the neuroscientific contributions of this paper. From a machine learning perspective, this paper has insufficient details to assess both the experimental contributions and proposed formulation of motivation. It is unclear from the discussion of biological forms of motivation, and from the experimental elaboration of these ideas, that the proposed model of motivation is a novel contribution. For these reasons, I suggest a reject.”
>
> The authors had little choice but to shorten the descriptions to meet the eight-pages-max requirements imposed by the ICLR format. On the bright side, it resulted in “the excellent clarity of this paper”, as pointed out by the Reviewer #2. From the comments after the Reviewer #3 we conclude, that they also perfectly inferred (the most of) the details from the text. We however agree with the Reviewer #3 that technical details should be explicitly present in the paper. To this end, we extended our paper with the Methods appendix, featuring sufficient detail to reproduce our computational experiments. We hope that this new section of the paper helps the Reviewer #3 clarify their technical questions, and therefore “assess both the experimental contributions and proposed formulation of motivation”.
>
> The neuroscientific contribution of the paper, briefly speaking, is the following. In behavioral experiments, mice are often kept water-deprived, so that the reward upon task completion is lucrative. This approach relies on the assumption that the perceived value of a reward depends not only on its physical value, but also on external factors. Although the existence of such dependence – named the motivational salience – is known in psychology and has been modelled in neuroscience, it is still unclear how motivation affects the reward perception in the brain. To this end, the results of numerous behavioral experiments critically rely on the process that is not yet well understood.
>
> To bridge this gap, we propose a functional circuit in the brain that may modulate the reward perception using motivation. Specifically, we found two large groups of cells in the ventral pallidum: cells tuned to positive and negative motivation respectively. We built an RL model suggesting that these two groups of cells may be connected through a recurrent “push-pull” circuit (Fig. 7F), retaining the information about the upcoming reward based on the cue and motivation. Overall, our work yields important insights into the reward processing in the brain, and motivates future studies in the reward circuitry, thus contributing to rigorous interpretation of behavioral data.
>
> The Four Rooms Experiment:
>
> “The issue with this example is that the non-motivated agent does not have access to the information required to learn a value-function suitable to solve this problem. By not giving the motivation vector to non-motivated agent, the problem has become a partially observable problem, and the comparison is now between a partially observable and fully observable setting, rather than a commentary on the difference between learning with and without motivation.”
>
> The authors argue that in the Four Rooms experiment, the latent variable (motivation) has deterministic dynamics; therefore, an agent can learn to predict the exact reward on every iteration based on location, making the latent variable (motivation) unnecessary for the state formulation. In this regard, the environment is fully observable. Moreover, an agent is capable of learning such reward dynamics – as shown by the non-motivated outlier agent in the Fig. 2B. Yet, the rest of non-motivated agents were far less successful in maximizing the reward, as compared to the motivated agents under the same learning parameters.
>
> Using the arguments above, the authors prefer to attribute the boost in learning to motivation. However, we would like to acknowledge that, to the letter of POMDP definition (having a latent variable hidden from the agent), the Reviewer #3 is right in classifying the non-motivated setting as POMDP. To this end, we also would not mind saying that motivation converts POMDP to MDP to facilitate learning the optimal policies.
>
> “In places, the claims made go beyond the results presented. How do we know that the non-motivated network is engaging in a "non-motivated delay binge"?“
>
> The authors thank the Reviewer #3 for this catch. We should have stated explicitly that for each run, we displayed sequences of agent's locations to establish correspondence between policies and average reward rates. We included this statement in the newly written Methods appendix.

---

> > ### Author Response · Authors · 2019-11-11
> > **Official Response to the Blind Review #3 - part 2**
> >
> > “ A single trial is insufficient to draw conclusions about the behaviour of an agent.”
> >
> > The Reviewer #3 has correctly inferred that each point on the Fig. 2B represents a single trial (3 types of agents at 41 values of θ). The authors agree that “a single trial is insufficient to draw conclusions about the behavior of an agent” if we consider specifically the behavior at the given set of conditions (e.g. fixed θ). In our task however, we iterate the values of θ with such step that the model performance under the adjacent values of θ is similar. To this end, single trials both allow for approximation of r(θ), and are indicative of the trial-to-trial performance variance. Just to be clear: there is nothing simpler than re-running the existing code ten times per model – and we did it – but the resulting plot becomes crowded. At the same time, not much information is added – the performance is highly conserved across re-runs – and the outliers, if any, are lost in case of averaging. Therefore, for now, the authors prefer to keep the plot as is, although we would appreciate further suggestions on it.
> >
> > The Path Routing Experiment:
> >
> > “In the second experiment, where a population of agents is presented in fig 5, it is claimed that on 82% of the trials, the agent was able to find the shortest path. Looking at the figure itself, at the final depicted iteration, all of the points are presented in a different colour and labelled “shortest path”. The graph suggests that 100% of the agents found the shortest path. The claim is made that for the remaining 18% of the agents, the agents found close to the shortest path—a point not evident in the figures presented.”
> >
> > The authors highly value this comment, as it highlights a source of confusion in the figure. Different color (red) represents the precomputed shortest paths, rather than the final depicted iteration. To alleviate this ambiguity, the new caption for the Fig. 5B reads: “(B) Test performance on 50 sets of 3 random targets (blue). The precomputed shortest path solutions (red).” Our comparison therefore reflects the differences between the last set of the blue points, and the set of the red points. The similarity between the blue and the red sets of points indicates that, despite 18% of the test simulations having not yielded the shortest path, the model’s path is always close to the shortest path.
> >
> > Pavlovian Conditioning Experiment:
> >
> > “In the third experiment, shouldn’t Q(s) be V(s)? In this setting, the agent is not learning the value of a state action pair, but rather the value of a state. Moreover, the value is described as Q(t), where t is the time-step in the trial; however, elsewhere in the text it is mentioned that the state is not simply t, but contains also the motivation value mu.”
> >
> > The authors have corrected the erroneous notation, thanks for the catch. It is important to write it as the Reviewer #3 suggested, i.e. V_t(s_t, μ_t), to indicate that the value function at the time t depends on the state and motivation. The time step t is not an explicit parameter of the value function V, and only affects it indirectly through the RNN. Therefore, our state construction is similar to the conventional one, except for the motivation.
> >
> > “The third experiment does not have enough detail to interpret the results. It is unclear how many trials there were for both of the prediction settings. It is unclear whether the problem described is a continuing problem or a terminating prediction problem—i.e., whether after the conditioned stimulus and unconditioned stimulus are presented to the agent, does the time-step (and thus the state) reset to 0, or does time continue incrementing?”
> >
> > It is a terminating prediction problem, as the Reviewer #3 has correctly guessed in the next comment – although we never mentioned it explicitly. To address this ambiguity, we included the following line in the Methods appendix: “We trained the network on terminating sequences of 20 iterations, representing time within individual trials”.
> >
> > “If it is a terminating prediction problem, it is unclear whether the conditioned stimulus and unconditioned stimulus were delivered on the same time-steps for each independent trial.”
> >
> > It is true that “the conditioned stimulus and unconditioned stimulus were delivered on the same time-steps for each independent trial” – although we never mentioned it in the text. To alleviate this uncertainty, we included the following lines in the Methods appendix: “Depending on iteration, the state variable was equal either to the CS (iterations 6-9 out of 20), or to zero (elsewhere)”; “Depending on iteration, the reward was equal either to the CS (iterations 15-16), or to zero (elsewhere)”. In other simulations (not included in the paper), we jittered the timings of the CS and the US (by 1, 2, or 3 time points out of 20), yet it did not affect any of our results. Thus, although in our simulations we fixed the CS and the US timings, they could as well be floating.

---

> > > ### Author Response · Authors · 2019-11-11
> > > **Official Response to the Blind Review #3 - part 3**
> > >
> > > “If I am interpreting the state-construction correctly, the state is incrementing by one on each time-step; this problem is effectively a Markov Reward Process where the agent transitions from one state to the next until time stops with no ability to transition to previous states. <…> In both the terminating and continuing cases, the choice of inputs is unusual. What was the motivation for using the time-step as part of the state construction?”
> > >
> > > The authors apologize for creating this confusion through ambiguous notation. The inputs that we use were the state and the motivation. We did not use the time step explicitly in the definition of the state – although it still has some effect on the value function through RNN, and through the dependence of the state on the time-step. The exact training details are now included in the Methods appendix: “On the input, the network received an agent’s state and motivation. The state variable contained a cue (conditioned stimulus; CS), which we chose randomly from {±0.0, ±0.4, ±0.8}. Depending on iteration, the state variable was equal either to the CS (iterations 6-9 out of 20), or to zero (elsewhere). The motivation variable μ = ±1 was equal to the sign of the CS; it was constant throughout the entire sequence of 20 iterations.” To further alleviate the confusion, we also used the new notation V_t(s_t, μ_t), as suggested by the Reviewer #3. This way, it becomes clear that our choice of the inputs is usual (except for the motivation), and the problem is a Markov Reward Process, which allows transitioning to the previously visited states.
> > >
> > > “How is the conditioned stimulus formulated in this setting? It is mentioned that it is a function of time, but there are no additional details.”
> > >
> > > We included the following line in the Methods appendix: “Depending on iteration, the state variable was equal either to the CS (iterations 6-9 out of 20), or to zero (elsewhere).”
> > >
> > > “From reading the text, it is unclear whether fig 7b/c presents activations over multiple independent trials or a single trial.”
> > >
> > > Those are activations in a single trial (one example per condition; 6 trials total). To address the ambiguity, the caption for the Fig. 7B,C now reads: “(B) Inputs and outputs of the RNN for each trial type. Inputs: motivation, cue, reward. Outputs: sample V-function. Bottom row: TD-error. Trial types (left to right): large negative reward, small negative reward, no negative reward, no positive reward, small positive reward, large positive reward. (C) Sample responses of neurons in the RNN clustered into the negative motivation (blue), positive motivation (red) and neurons of mixed sensitivity (other)”. Although the results presented on the Fig. 7B,C is highly conserved across the restarts, it would have been difficult to plot the average-case activations, as there is no unique mapping between the neurons in the model across multiple restarts. We therefore can only show the results for a single representative model.
> > >
> > > General Thoughts on Framing:
> > >
> > > “This paper introduces non-standard terms without defining them first. For example, TD error is introduced as Reward Prediction Error, or RPE: a term that is not typically used in the Reinforcement Learning literature.”
> > >
> > > As suggested by the Reviewer #3, we converted all the instances of RPE to the conventional notation: TD error
> > >
> > > “To my understanding, there is a hypothesis about RPE in the brain in the cognitive science community; however, the connection between this idea in the cognitive science literature and its relation to RL methods is not immediately clear.”
> > >
> > > The authors would like to thank the Reviewer #3 for this catch. We had the reference in the draft version of the introduction, but then it got lost in course of the edits. Now, then we switched to the conventional RL notation (TD error) the reference is not needed anymore.
> > >
> > > “Temporal Difference learning is incorrectly referred to as "Time Difference" learning (pg 2).”
> > >
> > > 	We fixed this typo, thanks.
> > >
> > > Notes on technical details:
> > >
> > > “The discounting function γ should be 0<= γ <=1, rather than just <=1.”
> > >
> > > 	We changed it as the Reviewer #3 suggested, although the original statement was also correct.
> > >
> > > “discounting not only prevents the sum of future rewards from diverging, but also plays an important role in determining the behaviour of an agent---i.e., the preference for short-term versus long-term rewards.”
> > >
> > > The authors agree with the Reviewer #3 that it is important to mention the role of the discounting factor in determining the behavior of an agent. It is especially important for the Transport network task, where an agent in 18% of the cases picks longer routes due to the larger expected discounted reward (analysis not shown). To signify the behavioral effect of the discounting factor, we add the following line: “Here γ is the discounting factor that keeps the sum from diverging, and balances preference of short- versus long-term rewards.”

---

> > > > ### Author Response · Authors · 2019-11-11
> > > > **Official Response to the Blind Review #3 - part 4**
> > > >
> > > > “pg 2 "the motivation is a slowly changing variable, that is not affected substantially by an average action" -- it is not clear from the context what an average action is.”
> > > >
> > > > To avoid ambiguity, the authors rewrote the sentence as: “The motivation is a slowly changing variable, that on average is not affected substantially by a single action.” Formally speaking, the average absolute value of time derivative of any component of the motivation vector is much less than the average absolute value of the same component of the motivation vector.
> > > >
> > > > “Why is the reward r(s|a), as opposed to r(s,a)?”
> > > >
> > > > The authors use this notation in sums of the expected future rewards to signify their dependence on the current action through the dependence of the future states and motivations on the current action. Perhaps, it would be correct and less ambiguous to write r_{t + τ}(s_{t + τ}, μ_{t + τ}, a_{t + τ} | a_t) ? If the Reviewer #3 suggests such change, we would be happy to introduce this edit in the upcoming version of the manuscript.
> > > >
> > > > Notes on paper structure:
> > > >
> > > > “There are some odd choices in the structure of this paper. For instance, the second section---before the mathematical framing of the paper has been presented---is the results section.”
> > > >
> > > > The authors argue that this seemingly unconventional choice is typical for the methods papers: as the mathematical framing of motivation is one of our contributions to RL, we present it in the Results section.
> > > >
> > > > “In some sentences, citations are added where no claim is being made; it is not clear what the relevance of the citation is, or what the citation is supporting. E.g., “We chose to use a recurrent neural network (RNN) as a basis for our model” following with a citation for Sutton & Barto, 1987.”
> > > >
> > > > We agree that this citation looks odd. The idea is that in 1978 paper, Sutton and Barto suggested using RNNs in the future research of Pavlovian conditioning – as we did. To make the reference more clear, though still to save space, we edited this line as follows: “we chose to use recurrent neural network (RNN) as a basis of our model, as suggested by Sutton & Barto (1987)”
> > > >
> > > > “In some sentences, citations are not added where substantial claims are being made. E.g, “The recurrent network structure in this Pavlovian conditioning is compatible with the conventional models of working memory”. This claim is made, but it is never made clear what the conventional computational models of working memory are, or how they fit into the computational approaches proposed.”
> > > >
> > > > Please find the references in the Results section: “During the delay, the network is supposed to maintain the information about upcoming reward, and, thus, acts as a working memory network (Her et al, 2016), which keeps reward expectation in its persistent activity. <…> Previous studies in working memory and decision-making tasks (Machens et al, 2005; Wong et al, 2007; Her et al, 2016) suggest that such parametric persistent activity can be maintained by two groups of oppositely tuned neurons, in the network architecture called the “push-pull” circuit.” We could not afford to duplicate the references due to the space constraints, so in parts of Discussion we simply reiterated the statements from the Results section.
> > > >
> > > > “Unfortunately, a number of readers in the machine learning community might be unfamiliar with pavlovian conditioning and classical conditioning. Taking the time to unpack these ideas and contextualise them for the audience might help readers understand the paper and its relevance.”
> > > >
> > > > We authors cannot agree more. Initially, we wrote up a comprehensive introduction to help Machine learning readers understand Neuroscientific relevance of the paper, and the other way around. Unfortunately, we had to shorten it substantially due to space constraints.
> > > >
> > > > “Figure 7B may benefit from displaying not just the predicted values V(s), but a plot of the prediction over time in comparison to the true expected return.”
> > > >
> > > > To evaluate the predicted values of V(s, μ), we plot the TD error (Fig. 7B, bottom row). The idea is similar to what the Reviewer #3 suggests; however, while the predicted and true V-functions are visually similar, TD error clearly displays where the true outcomes deviate from the expectations.

---

> > > > > ### Comment · AnonReviewer3 · 2019-11-15
> > > > > **Reply to Authors' Response**
> > > > >
> > > > > I acknowledge the authors' response and appreciate their diligence in responding to reviewer feedback.
> > > > >
> > > > > Given the aforementioned problematic formulation of the experiments and a lack of clarity in the paper's presentation, I maintain my recommended score.

---

> > > > > > ### Author Response · Authors · 2019-11-15
> > > > > > **Re: Reply to Authors' Response**
> > > > > >
> > > > > > We appreciate the reviewer’s comments.
> > > > > >
> > > > > > We would like to argue that the Reviewer #3 correctly understood most of the technical details from the original text. To increase clarity in the paper's presentation, we followed the Reviewer #3's recommendation and included all technical details in the revised manuscript, as well as we removed ambiguities in the main text. We also clearly addressed each of the Reviewer #3's concerns in the comments above.
> > > > > >
> > > > > > Since the rebuttal phase of the review process is intended for a discussion between the Authors and the Reviewers -- to clarify the details and revise the submission for the better-informed final valuation -- we would like to request the Reviewer #3 to kindly account for the comments above, and for the newly written methods appendix, in their final decision on the paper.

---

### Decision · Program_Chairs · 2019-12-19

**Decision:**

Reject

**Comment:**

This paper proposes a deep RL framework that incorporates motivation as input features, and is tested on 3 simplified domains, including one which is presented to rodents.

While R2 found the paper well-written and interesting to read, a common theme among reviewer comments is that it’s not clear what the main contribution is, as it seems to simultaneously be claiming a ML contribution (motivation as a feature input helps with certain tasks) as well as a neuroscientific contribution (their agent exhibited representations that clustered similarly to those in animals). In trying to do both, it’s perhaps doing both a disservice.

I think it’s commendable to try to bridge the fields of deep RL and neuroscience, and this is indeed an intriguing paper. However any such paper still needs to have a clear contribution. It seems that the ML contributions are too slight to be of general practical use, while the neuroscientific contributions are muddled somewhat. The authors several times mentioned the space constraints limiting their explanations. Perhaps this is an indication that they are trying to cover too much within one paper. I urge the authors to consider splitting it up into two separate works in order to give both the needed focus.

I also have some concerns about the results themselves. R1 and R3 both mentioned that the comparison between the non-motivated agent and the motivated agent wasn’t quite fair, since one is essentially only given partial information. It’s therefore not clear how we should be interpreting the performance difference. Second, why was the non-motivated agent not analyzed in the same way as the motivated agent for the Pavlovian task? Isn’t this a crucial comparison to make, if one wanted to argue that the motivational salience is key to reproducing the representational similarities of the animals?  (The new experiment with the random fixed weights is interesting, I would have liked to see those results.) For these reasons and the ones laid out in the extensive comments of the reviewers, I’m afraid I have to recommend reject.